# Introduction of a male-harming mitochondrial haplotype via 'Trojan Females' achieves population suppression in fruit flies

Jonci Nikolai Wolff[1]*, Neil J Gemmell[2], Daniel M Tompkins[3], Damian K Dowling[1]

[1]School of Biological Sciences, Monash University, Victoria, Australia; [2]Department of Anatomy, University of Otago, Dunedin, New Zealand; [3]Landcare Research, Dunedin, New Zealand

**Abstract** Pests are a global threat to biodiversity, ecosystem function, and human health. Pest control approaches are thus numerous, but their implementation costly, damaging to non-target species, and ineffective at low population densities. The Trojan Female Technique (TFT) is a prospective self-perpetuating control technique that is species-specific and predicted to be effective at low densities. The goal of the TFT is to harness naturally occurring mutations in the mitochondrial genome that impair male fertility while having no effect on females. Here, we provide proof-of-concept for the TFT, by showing that introduction of a male fertility-impairing mtDNA haplotype into replicated populations of *Drosophila melanogaster* causes numerical population suppression, with the magnitude of effect positively correlated with its frequency at trial inception. Further development of the TFT could lead to establishing a control strategy that overcomes limitations of conventional approaches, with broad applicability to invertebrate and vertebrate species, to control environmental and economic pests.

*For correspondence: jonci.
wolff@gmail.com

**Competing interests:** The authors declare that no competing interests exist.

## Introduction

Pest species pose some of the greatest present-day challenges to native biota, global economies, and human health (*Naranjo et al., 2015*; *Simberloff et al., 2013*). With their emergence and spread linked to human trade, transport and the agricultural revolution, vertebrate, invertebrate and plant pest impacts have followed human global movement over past millennia like a footprint (*Hulme, 2009*). The environmental and agricultural losses caused by pests have been estimated at US$120 billion annually in the US alone (*Pimentel et al., 2005*), while many also vector diseases of concern. For example, there are around 500 million human cases of the mosquito-borne diseases malaria and dengue fever annually (*World Health Organization, 2015*; *Bhatt et al., 2013*), with other agents such as Zika virus rapidly spreading (*World Health Organization, 2016*). Considering the humanitarian, economic, and environmental values at stake, pest management requires large and recurrent government expenditure worldwide (*Stenseth et al., 2003*).

Conventional pest management programs typically rely on some form of lethal control, such as vertebrate shooting, trapping, and poisoning, or the area-wide application of pesticides to target weeds and invertebrates. The effects of these approaches are often temporary in nature and thus require regular re-application, and may have unwanted side effects on non-target species and the environment (*Bergstrom et al., 2009*; *Tompkins and Veltman, 2006*; *Innes and Barker, 1999*). Furthermore, vertebrate culling is generally cost- and labor-intensive and can prove ineffective at low population densities (*Clout and Russell, 2006*), while blanket pesticide application is often

**eLife digest** Insect and other animal pests pose some of the greatest challenges to biodiversity, global economies and human health. The environmental and agricultural losses caused by pests have been estimated at 120 billion US dollars a year in the US alone. Many pests also spread diseases, such as dengue fever and malaria. A variety of different strategies are used to control pests, but their effects are generally short-lived and they are often ineffective when pest numbers are low. Furthermore, many of these strategies are harmful to other wildlife, such as bees.

Most of the DNA within an animal cell is contained within a structure called the nucleus. However, some DNA is also found within other compartments called mitochondria. The Trojan Female technique has been proposed as a new strategy to control insect pests that harnesses naturally-occurring changes (known as mutations) in this mitochondrial DNA (or mtDNA for short). Introducing mutations that lower the fertility of males, but have no effects on females, into a pest population should, in theory, lead to a long-lasting decline in the size of the population, even if it is relatively small to begin with.

Wolff et al. tested this theory in fruit flies, which are often used as models of insects and other animals in research projects. Adding female fruit flies carrying a mutation in mtDNA that lowers male fertility (known as "Trojan Females") into populations of fruit flies reduced the size of the population over several generations. The mutation was maintained in the population for at least ten generations, and no "rescue" mutations evolved in the nuclear DNA to compensate for the mtDNA mutation. This indicates that the Trojan Female technique could be effective at controlling pests, without the need for Trojan Females to be repeatedly released into the populations.

The next steps following on from this work are to test this approach in economically important pest species, and to find out whether the approach is effective in various environments outside the laboratory. If these findings do indeed translate into these pests, then the Trojan Female technique may have the potential to be used to control a wide variety of different pest species from mosquitos through to rats.

hampered by the evolution of resistance in target species (*Tabashnik et al., 2008*; *Innes and Barker, 1999*). Research efforts have thus focused on the development of novel control or eradication techniques that are target-specific, cost-effective even when applied at low population densities, and long-lasting in effect.

One promising avenue lies in the development of techniques that impair the reproductive capacity of target pest species (*Cowan et al., 2002*; *Courchamp and Cornell, 2000*). The most successful of such techniques employed to date is the Sterile Insect Technique (SIT), whereby large quantities of sterilized males are introduced into target populations, reducing the reproductive success of the females with whom they mate (*Alphey et al., 2010*). Although the SIT offers species-specificity, a major constraint lies in the need to continuously produce and release large numbers of sterile males for sustained population suppression, rendering eradication efforts time- and cost-intensive (*Dyck et al., 2005*; *Alphey et al., 2010*). Resource requirements could be greatly reduced if impairments to male fertility in a target population were heritable in nature. Emerging theory and experimental work suggests this is achievable via a prospective approach called the Trojan Female Technique (TFT; *Gemmell et al., 2013*).

The goal of the TFT is to use naturally occurring mutations in the mitochondrial DNA (mtDNA), which impair male fertility but have no effects on females, to achieve multi-generational pest population suppression. Because mtDNA is typically maternally inherited (*White et al., 2008*; *Birky, 1978*), such male-specific deleterious mtDNA mutations will to a large degree escape selection in the female germ line despite their associated high fitness cost to males, enabling their persistent inheritance across generations (*Frank and Hurst, 1996*; *Beekman et al., 2014*). In theory, 'Trojan Females' carrying such mutations, and their female descendants, could continuously produce males with impaired fertility across generations, achieving perpetual numerical suppression of target populations (*Gemmell et al., 2013*). Unlike other genetically based pest control approaches that involve transgenics, such as the Release of Insects carrying a Dominant Lethal (RIDL) (*Thomas et al., 2000*),

and the theorized use of gene-drives (*Webber et al., 2015*; *Taylor and Gemmell, 2016*), the use of naturally occurring mutations means that TFT pest control would not necessarily require genome editing to progress. The TFT may thus offer a valuable alternative to emerging transgenic control techniques, whose potential use is currently subject to debates concerning safety and regulatory concerns (*Oye et al., 2014*; *Lunshof, 2015*).

The conceptual framework underpinning the TFT is based on a population genetic model that shows the maternal inheritance of mitochondria will facilitate the accumulation of deleterious mtDNA mutations that are male-biased in their effects (*Gemmell et al., 2004*; *Frank and Hurst, 1996*; *Beekman et al., 2014*). Recent empirical work in *Drosophila melanogaster* has substantiated this model, showing that the expression of fertility, longevity, and levels of nuclear gene expression are more sensitive to genetic variation in the mtDNA sequence in males than in females (*Camus et al., 2012*; *Yee et al., 2013*; *Innocenti et al., 2011*; *Camus et al., 2015*). Furthermore, particular mtDNA haplotypes have now been associated with sub- or complete-infertility in males, but with no apparent effects on female fertility, in *Drosophila* (*Dowling et al., 2015*; *Yee et al., 2013*; *Patel et al., 2016*; *Wolff et al., 2016b*), seed beetles (*Dowling et al., 2007*), hares (*Smith et al., 2010*), and humans (*Ruiz-Pesini et al., 2000*). Based on this theory and empiricism, *Gemmell et al., 2013* explored the conditions under which TFT mutations (male-fertility impairing but female-benign) could lead to population suppression. The results were encouraging, indicating that TFT haplotypes are predicted to cause suppression across a wide range of life-histories, with such effects not only persisting across generations, but also accumulating across successive introductions (*Gemmell et al., 2013*).

Building on those initial models (*Gemmell et al., 2013*), empirical attention has focused on a particular mitochondrial haplotype sourced from a population of *D. melanogaster* in Brownsville (USA), which has been associated with perturbed spermatogenesis and sperm maturation (*Clancy et al., 2011*) and is known to confer complete male sterility when placed alongside one isogenic nuclear background (i.e. a nuclear background devoid of any allelic variation; *Clancy et al., 2011*), and consistent reductions in male fertility against a range of other nuclear backgrounds (*Dowling et al., 2015*; *Yee et al., 2013*; *Wolff et al., 2016b*). Sequence analyses revealed 10 single nucleotide polymorphisms (SNPs) that are unique to this Brownsville haplotype: one mutation located in the cytochrome b gene and nine others that reside within the A/T-rich control region (mt:Cyt-b; *Wolff et al., 2016a*). The SNP located in the mt:Cyt-b gene is non-synonymous, causing an amino acid change (Ala278→Thr) in complex III of the mitochondrial electron transfer chain, and it is this SNP that has previously been implicated as the putative fertility-reducing mutation (*Camus et al., 2015*; *Clancy et al., 2011*). This mt:Cyt-b SNP as the cause of the mtDNA-mediated male infertility has yet to be unambiguously confirmed; such confirmation would require further work to disassociate this mutation from the other nine SNPs that delineate the Brownsville haplotype from its counterparts, which would most likely be tractably accomplished by gene editing – an approach that remains in its infancy for the mitochondrial genome (*Wisnovsky et al., 2016*).

A recent study has provided further support for a key role for this SNP in fertility suppression (*Camus et al., 2015*). Camus *et al.* (2015) demonstrated that the gene in which the SNP lies (mt:Cyt-b) experiences a four-fold decrease in expression in flies carrying the Brownsville haplotype relative to flies with other haplotypes, while expression of other mtDNA protein-coding genes is unaffected. Intriguingly, this Ala278→Thr mutation in the mt:Cyt-b gene occurs naturally in a range of other species, both vertebrate and invertebrate (*Clancy et al., 2011*). Although the phenotypic implications of this mutation have not yet been screened outside of its putative effect in *D. melanogaster*, this indicates that male-fertility-reducing mtDNA haplotypes may routinely segregate in natural populations of metazoans (*Frank and Hurst, 1996*; *Beekman et al., 2014*; *Gemmell et al., 2004*). However, the practical utility of harnessing male-fertility-reducing haplotypes for pest control remains unclear on two fronts: first, whether the reductions in male fertility that they cause will indeed result in the numerical suppression of populations and second, whether any demonstrable suppression effects will persist across generations.

There are several mechanisms by which impaired fertility in individual males may be compensated for at both individual and population scales. First, even males with impaired fertility may provide sufficient viable sperm for the complete fertilization of the eggs of females with which they mate in a population context. Second, even if a female is viable-sperm limited when mated with an impaired male, she may still obtain sufficient viable sperm through mating with other males. Third, even if

population suppression initially occurs, as yet undetected pleiotropic effects on females could select against the mutation across generations. Fourth, even if the mutation persisted, the selection pressure imposed on males could select for nuclear modifiers that compensate for the effect of the TFT mitochondrial haplotype. Evolutionary theory and empiricism suggest that fertility effects associated with male-harming mtDNA mutations will often be reduced in such a way (*Yee et al., 2013*; *Frank and Hurst, 1996*; *Gemmell et al., 2004*). Ultimately, when present in large and panmictic populations, TFT haplotypes may be expected to undergo changes in frequency through genetic drift and directional or balancing selection (*Wolff et al., 2014*; *Gregorius and Ross, 1984*; *Clark, 1984*). Furthermore, stochastic contractions and expansions in the target population may exacerbate the effects of genetic drift and facilitate the purging of introduced TFT haplotypes, even when they are not selected against (*White et al., 2008*; *Rand et al., 2001*).

Here, we experimentally test the capacity of the Brownsville haplotype (our candidate TFT haplotype) to suppress large and panmictic laboratory populations of *D. melanogaster*. Persistent numerical suppression would provide proof-of-concept for the TFT, showing that the maternal mode of mtDNA inheritance can potentially be harnessed for a eukaryotic pest control technique that overcomes several limitations of conventional approaches. Trial populations were initiated with the TFT haplotype at four different starting frequencies (0% [control], 25%, 50%, and 75%), with the expectation that the numerical suppression observed would increase with increasing TFT frequency, and these populations were then maintained for 10 generations under two environmental regimes.

Under the first regime, populations were maintained in the ecological and demographic conditions in which they are typically maintained in the laboratory, and in which egg numbers per generation are carefully regulated. Predictions from an existing simulation model of *D. melanogaster* populations under such a regime (see Supplementary information in [*Wolff et al., 2016b*]) are that the TFT haplotype utilized (documented to reduce male breeding success by 29–69%; *Dowling et al., 2015*; *Wolff et al., 2016b*) will cause mean population suppression of between 6.7–16.8%, 13.8–37.7% and 21.4–63.7% for the 25%, 50% and 75% TFT haplotype frequency, respectively. Such a magnitude of suppression, modeled under conditions of multiple mating, is predicted to translate into up to three times greater suppression in natural populations in which females remate at a relatively low frequency (*Wolff et al., 2016b*). Under the second regime, populations were maintained in conditions that allow them to experience stochastic contractions and expansions in population size that are more reflective of natural population dynamics. The purpose of this regime was to explore how such dynamics could influence the frequency of the TFT haplotype across generations, with genetic drift potentially leading to either haplotype purging (with associated loss of population suppressive effects) or fixation (with associated increase in population suppressive effects).

Together, our experiments document the first experimental test of the ability of a candidate TFT haplotype to cause and maintain population suppression. We demonstrate that the TFT haplotype caused significant numerical suppression in the laboratory *Drosophila melanogaster* populations relative to controls, with the magnitude of effect positively correlated with its frequency at trial inception. Furthermore, the suppressive effect persisted over the full length of the trial (10 generations), with no reduction in haplotype frequency. Our results thus provide proof-of-concept for the TFT, showing that uniparental inheritance of mtDNA could potentially be harnessed in the development of a pest control technique that would be broadly relevant across eukaryotes.

## Results

### Experiment 1 – population suppression under density-controlled conditions (regulated population size)

We found an interactive effect of TFT treatment and generation number on population sizes (*Tables 1* and *2A*), and on the frequency of the TFT mutation (*Tables 1* and *2B*), across the 10 generations of the experiment (*Figure 1A*). Replicates initiated with TFT haplotype frequencies of 0% or 25% stabilized at average population sizes of 72.99 and 72.57 individuals across the 10 generations, respectively. However, replicates initiated with TFT haplotype frequencies of 50% or 75% declined over the first six generations to average population sizes of 67.24 and 66.75 respectively, with this magnitude of suppression (8%) maintained for the remainder of the experiment.

**Table 1.** Mean offspring numbers, TFT haplotype frequencies, and genotyping outcomes for the two experiments (regulated conditions and fluctuating conditions).

| | | TFT treatment (starting frequency) | 0% TFT | 25% TFT | 50% TFT | 75% TFT |
|---|---|---|---|---|---|---|
| | | Populations [n] | 21 | 21 | 21 | 21 |
| Experiment | Regulated | $F_1$ Frequency | 0.00 | 0.26 ± 0.03 | 0.44 ± 0.03 | 0.63 ± 0.03 |
| | | $F_5$ Frequency | 0.00 | 0.15 ± 0.03 | 0.63 ± 0.03 | 0.71 ± 0.02 |
| | | $F_{10}$ Frequency | 0.00 | 0.17 ± 0.03 | 0.67 ± 0.03 | 0.80 ± 0.03 |
| | | Loss | - | 4 | 0 | 0 |
| | | Fixation | - | 0 | 0 | 3 |
| | | Heteroplasmy | - | 0 | 0 | 0 |
| | | Mean offspring number ($F_{10}$) | 74.15 ± 1.00 | 72.05 ± 0.89 | 67.43 ± 0.99 | 66.38 ± 0.70 |
| | | Population extinction | 1 | 0 | 0 | 0 |
| | Fluctuating | $F_{10}$ frequency | 0.00 | 0.35 ± 0.06 | 0.59 ± 0.06 | 0.75 ± 0.05 |
| | | Loss | - | 4 | 1 | 0 |
| | | Fixation | - | 1 | 4 | 4 |
| | | Heteroplasmy | - | 0 | 2 | 0 |
| | | Mean offspring number ($F_{10}$) | 80.47 ± 5.12 | 81.19 ± 4.59 | 78.56 ± 5.63 | 85.86 ± 3.73 |
| | | Population extinction | 2 | 0 | 2 | 0 |

Source data 1. Raw data for offspring number and TFT frequency for Experiments 1 and 2. Offspring number per population for each of 10 generations in Experiment 1 and 2. TFT frequency for each population at generations 1, 5 and 10 for Experiment 1 and at generation 10 for Experiment 2.

We genotyped females from each experimental population at generations 1, 5 and 10, and found that the population suppression effect across generations was associated with the frequency of the TFT haplotype ($\chi^2$=21.37, p=<0.003; *Table 3*, *Figure 2A–C*). There was no evidence of consistent TFT haplotype purging; while the 25% TFT treatment ended (at generation 10) at a mean haplotype frequency of 0.17, ending frequencies were 0.67 and 0.80 for the 50% and 75% TFT treatments

**Table 2.** (A) Linear mixed model showing effects of TFT treatment and generation number on mean offspring number, and (B) generalized linear mixed model of effects of TFT treatment and generation on TFT haplotype frequency of populations with regulated population size (Experiment 1). There was no evidence of overdispersion in the model of TFT haplotype frequency (dispersion parameter = 0.766), and addition of an observation-level random effect to the final model did not change this parameter (dispersion parameter = 0.767), nor the parameter estimates of the model.

| | (A) Offspring number | | | (B) TFT frequency | | |
|---|---|---|---|---|---|---|
| *Fixed effects* | $\chi^2$ | Df | p | $\chi^2$ | Df | p |
| TFT treatment | 2.23 | 3 | 0.53 | 38.14 | 2 | <0.001 |
| Generation | 14.73 | 9 | 0.10 | 6.21 | 2 | 0.045 |
| TFT treatment × generation | 51.09 | 27 | 0.003 | 23.52 | 4 | <0.001 |
| *Random effects* | SD* | | | SD* | | |
| Biological replicate | 0.81 | - | - | 0 | - | - |
| Experimental population | 1.11 | - | - | 0 | - | - |
| Residual | 5.48 | - | - | - | - | - |
| * Standard deviation | | | | | | |

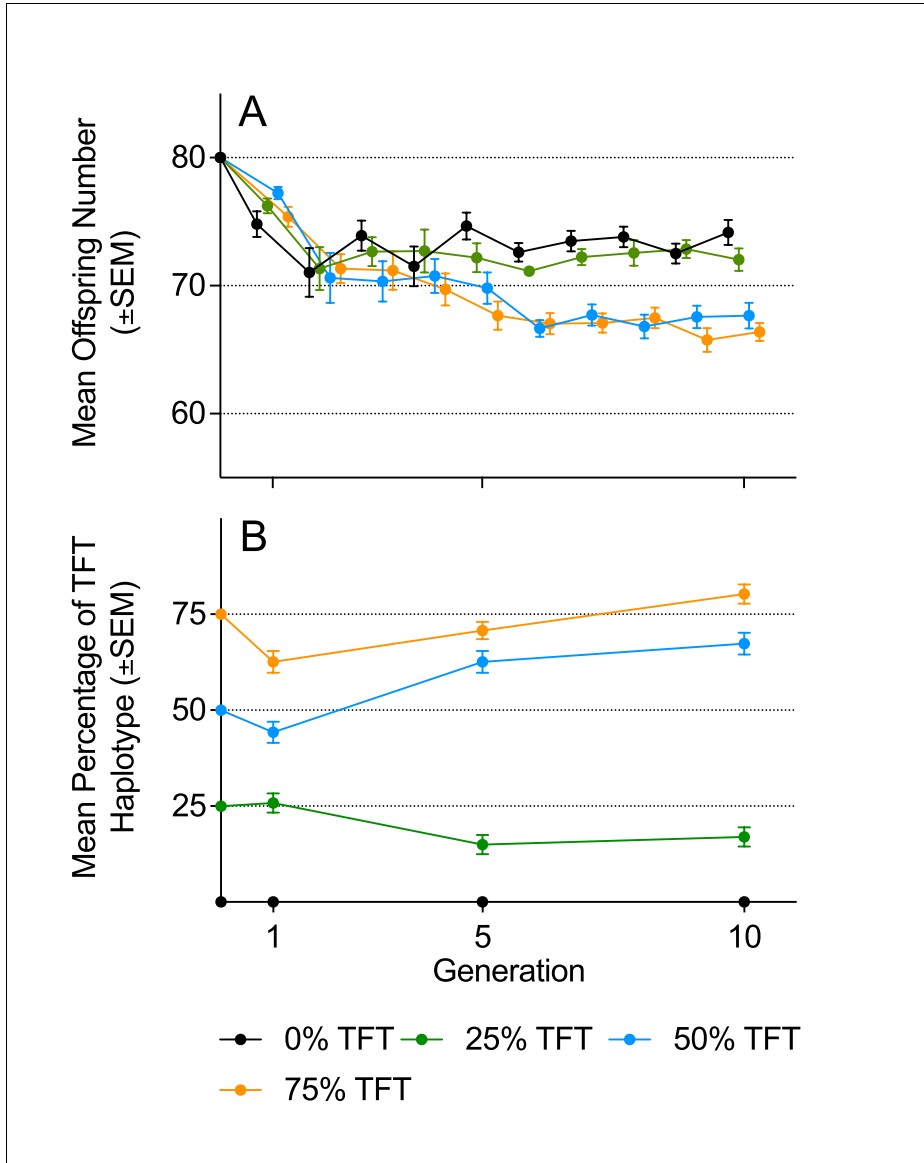

**Figure 1.** Mean offspring number and haplotype frequencies under density-controlled population conditions (Experiment 1). (**A**) Mean offspring number (±SEM), and B) mean haplotype frequencies (±SEM) of experimental populations in Experiment 1 over 10 generations, under density-controlled conditions. Founding populations ($F_0$) were established with varying proportions (0%, 25%, 50%, 75%) of the TFT haplotype. Each generation was propagated with 80 eggs. Genotyping was conducted at generations 1, 5, and 10.

The following source data is available for figure 1:

**Source data 1.** Raw data for offspring number and TFT frequency for Experiment 1.

respectively (*Figure 1B*; *Table 1*). Over the course of the trial, the TFT haplotype was purged from four replicates and went to fixation in three (out of total of 63 populations).

## Experiment 2 – population suppression under stochastic dynamics (Fluctuating population size)

The less-constraining rearing conditions of this experiment led to mean changes in population size between any two generations of approximately 30% across all experimental populations ($\chi^2$=46.72, p=<0.001; *Figure 3*; *Table 4*), with four experimental populations (two in each of the 0% and 50%

**Table 3.** Linear mixed model of association of TFT haplotype frequency on mean offspring number in populations with regulated population size (Experiment 1).

| Fixed effects | $\chi^2$ | Df | P |
|---|---|---|---|
| TFT frequency | 21.37 | 7 | 0.003 |
| Random effects | SD* | | |
| Biological replicate | 0.31 | - | - |
| Experimental population | 0 | - | - |
| Generation | 2.77 | - | - |
| Residual | 6.40 | - | - |

* Standard deviation

TFT treatments) going extinct. Under these conditions, there was no detectable effect of the TFT treatment on population size across the experiment (*Table 4A*). Nor did we detect an association between the TFT haplotype frequency of each experimental population at generation 10 and the final population size (*Table 5*, *Figure 2D*). However, the TFT haplotype was still stably maintained in most cases; ending frequencies were 0.35, 0.59 and 0.75 for the 25%, 50% and 75% TFT treatments, respectively (*Table 1*, *Table 4B*, *Figure 3*). Over the course of the trial, the TFT haplotype was lost from five replicates and went to fixation in nine (out of a total of 63 populations). In two replicates of the 50% TFT treatment, we detected 11 cases where flies carried both the TFT and wildtype haplotype (five individuals in one replicate and six in the other), indicating at least two cases of biparental mtDNA inheritance in the experiment.

## Discussion

Working with laboratory populations of *D. melanogaster*, we have demonstrated that the compromised male fertility caused by our candidate TFT haplotype can suppress population sizes across generations. The magnitude of suppressive effect was dependent on the frequency of the TFT haplotype and the conditions under which populations were propagated. When the experiment was conducted under regulated population sizes, persistent numerical suppression was observed (*Figure 1A*). While all treatments reduced in population size over the first two generations, as the trial moved toward equilibrium dynamics, those seeded with at least 50% TFT haplotypes continued to decline to generation six, and then remained suppressed at sizes averaging 8% below control populations. No such effect was apparent for populations seeded with only 25% TFT haplotypes.

These results raise two questions. First, why was the suppressive effect only 8% at the population scale, when a-priori modelling predicted suppression relative to controls of at least 21.4% in the 75% TFT haplotype frequency treatment (see Supplementary information in *Wolff et al., 2016b*). Second, why was there no apparent effect of the 25% TFT treatment? These two issues are likely linked, and may be due to compensation at the scale of the individual, through females obtaining more fertile sperm than would be expected on an additive basis, either wholly from the sub-fertile TFT males with whom they mate and/or through multiple mating (i.e. the TFT haplotype used may have caused less reduction in male breeding success than modelled, or females may have mated with more males than included in our modelling (see Supplementary information in *Wolff et al., 2016b*; *Gemmell et al., 2013*)). Such compensation would explain why population size was not affected in the 25% TFT treatment (complete compensation) and was less than predicted in the 50% and 75% TFT treatments (partial compensation). The reduction in haplotype frequency that occurred in the 25% TFT treatment (*Figure 1B*; *Table 1*) may also have contributed. However, irrespective of these or other mechanisms that may be responsible, a significant population suppression effect of the TFT haplotype, which persisted to the end of the trial, was observed for populations of the 50% and 75% TFT treatments.

When the experiment was conducted under conditions in which populations experienced large stochastic contractions and expansions in size, no suppression effects were detected. This was not driven by overall changes in TFT haplotype frequency over the course of the trial. While, as was

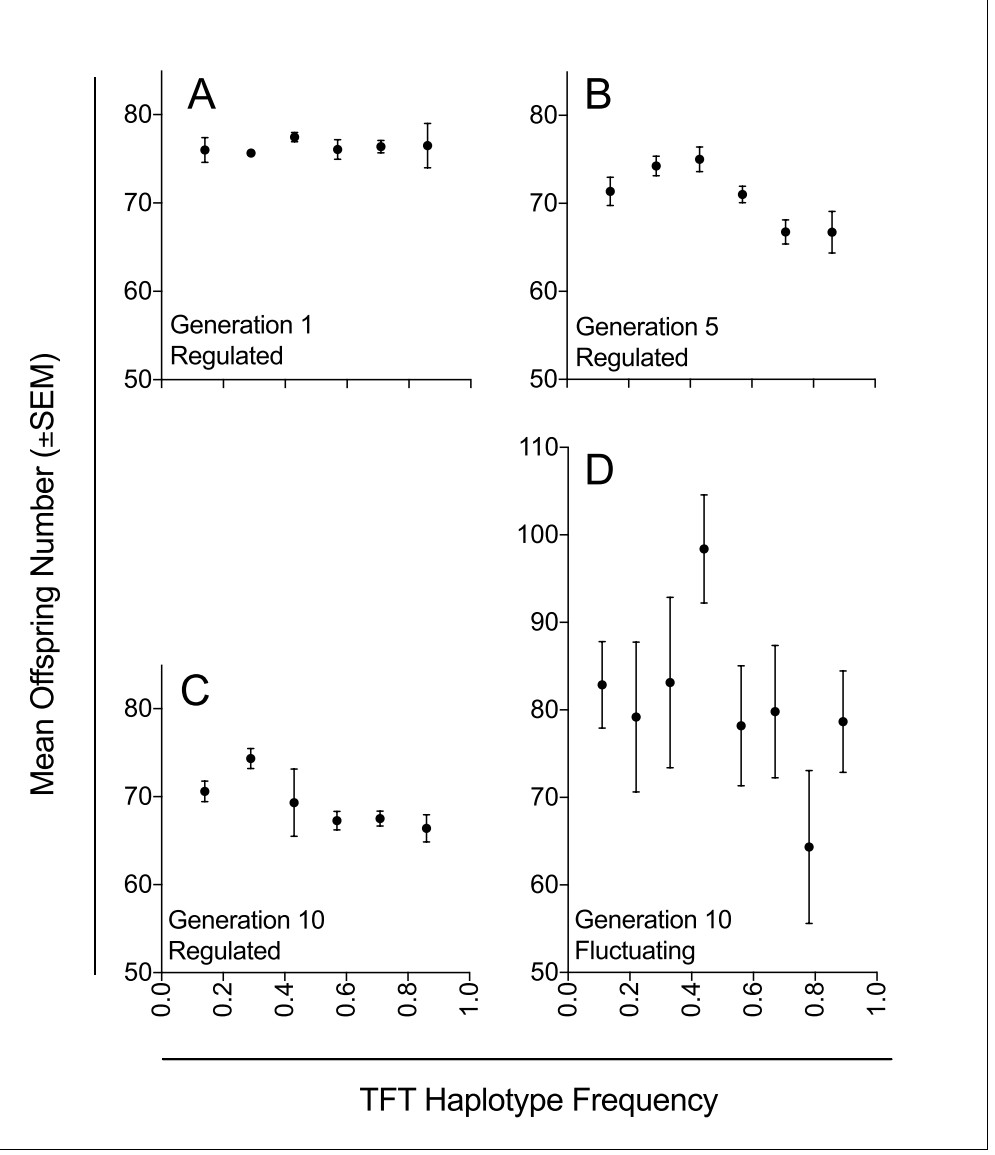

**Figure 2.** Mean offspring number in relation to TFT haplotype frequency under density-controlled (Experiment 1) and fluctuating population conditions (Experiment 2). Mean offspring number (±SEM) across experimental populations (A–C) with regulated population size at generations 1, 5, and 10 in Experiment 1; and (D) with fluctuating population size at generation 10 in Experiment 2. Haplotype frequencies were determined by genotyping seven females for each experimental population at each of three generations (1, 5, and 10; n = 1323) in Experiment 1; and by genotyping nine females for each experimental population at generation 10 (n = 567) in Experiment 2.

The following source data is available for figure 2:

**Source data 1.** Raw data for offspring number and TFT frequency for Experiments 1 and 2.

expected, there were more cases of haplotype loss and fixation under this regime than under the more stable regime (totals of 14 versus 7 such events), the haplotype went to fixation more often than it was purged, and no overall decline in TFT frequency from starting conditions was observed (*Figure 3*). It is thus most likely that the suppression effect of around 8% observed under the more stable experimental regime, was masked by the underlying population dynamics in the more stochastic regime. Although mean TFT haplotype frequency declined slightly (from 25% to 17%) across

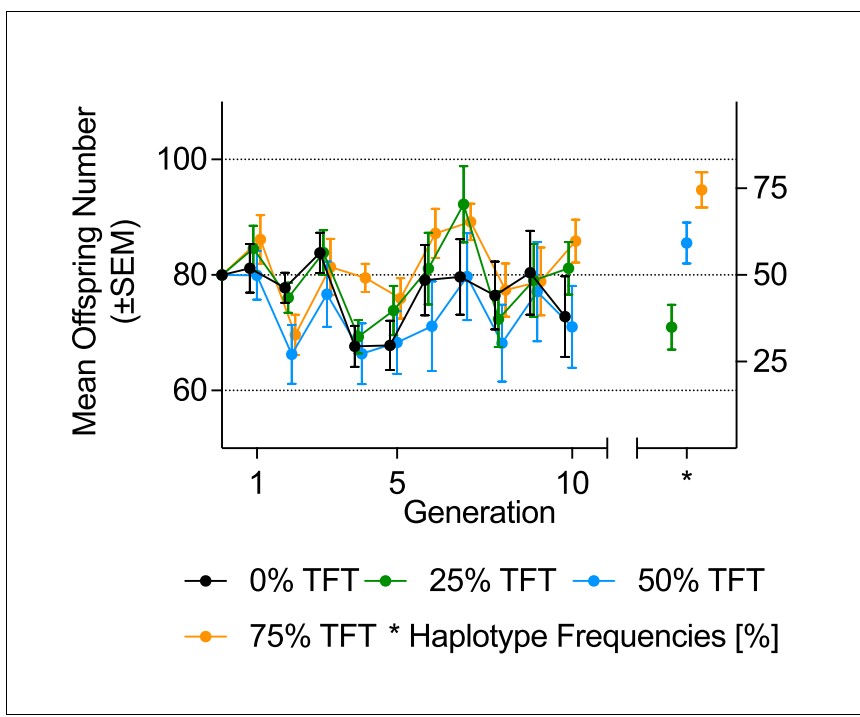

**Figure 3.** Mean offspring number and haplotype frequencies under fluctuating population conditions (Experiment 2). Mean offspring number (±SEM, on vertical axis on left-hand side), and mean haplotype frequencies (±SEM, right-hand side) at generation 10, of experimental populations. Founding populations ($F_0$) were established with varying proportions (0%, 25%, 50%, 75%) of fruit fly pairs harboring the mt:Cyt-b TFT mutation. Each generation, each experimental population was propagated with all offspring of the previous generation.

The following source data is available for figure 3:

**Source data 1.** Raw data for offspring number and TFT frequency for Experiment 2.

the 10 generations of Experiment 1 in the 25% TFT treatment, the frequency of the TFT haplotype actually increased in the 50% and 75% treatments (to an average of 67% and 80%, respectively; *Figure 1A–B*) of this experiment. Furthermore, in Experiment 2, haplotype frequencies increased in the 25% and 50% treatments and were stably maintained in the 75% treatment at generation 10. Thus, across six treatments and two experiments, the frequency of the TFT haplotype decreased in just one treatment, and this decrease was modest. These data suggest there was no strong selective pressure against the TFT haplotype, even though it was causing population suppression. This result further supports both the theory underpinning the TFT, that male-infertility caused by a mutation in the mtDNA will generally escape selection due to its maternal inheritance (*Frank and Hurst, 1996*; *Gemmell et al., 2004*; *Beekman et al., 2014*), and the previous anecdotal observations of no negative pleiotropic effects on female fertility being linked to the TFT haplotype (which would also incur selection against it; *Clancy et al., 2011*; *Dowling et al., 2015*; *Yee et al., 2013*). Intriguingly, the frequency of the TFT haplotype increased in four of the six treatments (*Table 1*). While this contention requires further experimental testing, if haplotype frequency increases are occurring one potential explanation may lie in an observation from a previous study that the TFT haplotype is linked to increased pupal viability (*Wolff et al., 2016b*), suggesting an antagonistic pleiotropic effect (low male fertility, but high pupal viability) that is under positive selection due to its benefits to females.

As noted in the introduction, persistent maintenance of TFT haplotypes within a population is expected to result in selection on males for nuclear modifiers that compensate for the negative TFT effect, and restore male fertility (*Frank and Hurst, 1996*; *Yee et al., 2013*; *Wolff et al., 2016b*; *Dowling et al., 2015*). The capacity for an effective compensatory response will depend largely on levels of standing nuclear allelic variance already present within populations. Our previous work

**Table 4.** (A) Linear mixed model showing effects of TFT treatment and generation on mean offspring number; and (B) generalized linear mixed model of effects of TFT treatment on TFT haplotype frequency of populations with fluctuating population size (Experiment 2). When reanalyzing mean offspring number (A), having excluded the 15 zero values in the dataset resulting from three vial extinctions (two from the TFT 0% treatment, and one from the TFT 50% treatment), the effect of TFT treatment on offspring number remained statistically non-significant ($\chi^2$=3.2, p=0.36). The binomial model of TFT frequency (b) indicated overdispersion (overdispersion parameter = 1.85), and thus an observation-level random effect was added (experimental population) to the model (overdispersion parameter of final model = 1.07).

| Fixed effects | (A) Offspring number | | | (B) TFT frequency | | |
|---|---|---|---|---|---|---|
| | $\chi^2$ | Df | P | 2 | Df | P |
| TFT treatment | 4.98 | 3 | 0.17 | 20.33 | 2 | <0.001 |
| Generation | 46.72 | 9 | <0.001 | - | - | - |
| Random effects | SD* | | | | | |
| Biological replicate | 3.1 | - | - | 0 | - | - |
| Experimental population | 11.2 | - | - | 1.32 | - | - |
| Residual | 20.85 | - | - | - | - | - |
| * Standard deviation | | | | | | |

indicated that although the effects of the TFT haplotype on male fertility consistently conferred lower fertility in males relative to other haplotypes, it was indeed modulated by the nuclear background of different populations (*Wolff et al., 2016b*; *Dowling et al., 2015*). However, even though the experimental populations utilized in the current study were large and expected to maintain high levels of segregating nuclear allelic variance (*Gardner et al., 2005*; *Griffin et al., 2016*), there was no apparent rapid selection for fertility-restoring nuclear components over the course of our trials (which would have been expressed as restoration in population sizes over time). Thus, although nuclear mutations could arise over time in a population that compensate for the negative effects of the TFT mutation, in the population of *Drosophila* that we used there were no apparent segregating nuclear modifiers that had the capacity to completely restore male fertility and be rapidly selected. Extending our experiments by placing the TFT haplotype alongside additional outbred nuclear backgrounds, including the nuclear background from the population the TFT haplotype was originally sourced from, could potentially inform at what frequency (if at all) such nuclear modifier alleles may occur. In this regard, it would also be interesting to evaluate whether the use of fertility-reducing mutations that have evolved naturally are likely to be more successful in the long-term (in terms of heritability and sustained effect) to suppress population size than the use of artificial gene-drive constructs, whose introduction are predicted to almost inevitably lead to the emergence of drive-resistant alleles in most natural populations (*Unckless et al., 2017*; *Noble et al., 2016*). Notably, if the suppression effect of these mutations (be they natural or gene-drive constructs) when placed into

**Table 5.** Linear model of association of TFT haplotype frequency on mean offspring number in populations with fluctuating population size (Experiment 2).

| Fixed effects | $\chi^2$ | Df | P |
|---|---|---|---|
| TFT Frequency | 4.95 | 9 | 0.839 |
| Random effects | SD* | | |
| Biological replicate | 0 | - | - |
| Residual | 24.93 | - | - |
| * Standard deviation | | | |

new target pest populations is large, this will presumably act to reduce both the efficacy by which selection can target standing nuclear variation, and the likelihood of spontaneous compensatory mutations, that restore fertility in the target population.

Interestingly, we identified cases in which offspring were heteroplasmic for both the TFT and Dahomey mtDNA haplotypes in two of the replicate populations. Heteroplasmy has previously been found in *Drosophila* sourced from Brownsville (*Kann et al., 1998*). Whether the Brownsville population is predisposed to sporadic episodes of biparental inheritance is unclear, but the repeated observation of low rates of biparental inheritance of mtDNA in populations across the globe suggests that paternal leakage may be common in *Drosophila* (*Wolff et al., 2013*; *Nunes et al., 2013*; *Dokianakis and Ladoukakis, 2014*). The intra-individual co-occurrence of both TFT and wildtype haplotypes enables the possibility for recombination between divergent mtDNA molecules to create novel mitochondrial haplotypes carrying the TFT mutation (*Ma and O'Farrell, 2015*). Whether the fertility-suppressing effect of the candidate TFT mutation(s) would be moderated by its placement alongside a different mitochondrial genetic background, within a recombinant haplotype, is unknown and would depend on the capacity for epistasis within the mitochondrial genome to affect fitness outcomes. Furthermore, to come into effect, such a novel recombinant haplotype must then be at a selective advantage if it is to become rapidly fixed within populations. However, a more likely scenario is that rare recombinant haplotypes will be purged under drift while segregating at low population frequencies, or under purifying selection if harmful to females (*Wolff et al., 2011*; *Bergstrom and Pritchard, 1998*; *Ma and O'Farrell, 2016*).

While we have provided proof-of-concept for the TFT, demonstrating experimentally that its male fertility effects can achieve persistent population suppression, the question remains of its utility for field application to pest populations. Critically, while the conceptual foundation for the work was based on TFT haplotypes conferring complete male sterility (*Clancy et al., 2011*), subsequent work has demonstrated that reduced fertility is the more likely scenario (*Dowling et al., 2015*; *Wolff et al., 2016b*), and our current laboratory trials indicate that compensation due to females remating with wild-type males, or other processes such as mitonuclear, or gene-by-environment interactions, could result in reduced levels of population suppression than would otherwise be predicted by the underpinning theory (*Gemmell et al., 2013*). However, modeling predicts the magnitude of effect to be much greater in natural populations in which females are expected to re-mate at a relatively low frequency (*Wolff et al., 2016b*), due to lower encounter rates between individuals of each sex and lower densities of cohabitation (*Gemmell et al., 2004*). Previous experiments have further revealed that individuals harboring the Brownsville haplotype exhibited increased pupal viability, which is likely to aid the introgression of the TFT haplotype into target populations (*Wolff et al., 2016b*).

The utility of the TFT in the field will also depend on the efficacy of specific TFT haplotypes to decrease male fertility (*Gemmell et al., 2013*). Multiple-release strategies can be employed in order to reach TFT haplotype frequencies required to achieve population suppression of natural populations. This way, eradication strategies may still be achievable even if the desired effect in population suppression is reliant on TFT haplotype frequencies that are high. In addition, there is potential to augment the sterilizing effects of the TFT haplotype through linking it with further candidate TFT mutations. Evolutionary theory and empiricism both suggest that plant and animal mitochondrial genomes should be naturally enriched for male-harming mtDNA mutations (*Innocenti et al., 2011*; *Gemmell et al., 2004*; *Frank and Hurst, 1996*; *Camus et al., 2012*; *Beekman et al., 2014*; *Dobler et al., 2014*). Indeed, a male-sterilizing but female-benign mutation has recently been discovered in the gene encoding the cytochrome c oxidase subunit 2 (mt:COII) in *D. melanogaster* (*Patel et al., 2016*). Linking multiple TFT mutations within a single TFT haplotype, or the release of multiple TFT strains each bearing a distinct set of TFT mutations holds great promise to further the capacity of the TFT to efficiently suppress population size. The pairing of multiple TFT mutations within the one mtDNA sequence should soon be quickly achievable, given the rapid advances in genome editing technologies (*Reddy et al., 2015*). Thus, although the TFT does not necessarily require transgenics to progress, genome editing could enable the time- and cost-efficient placement of TFT mutations into target populations (without the need for long-running mutagenic and breeding approaches to first generate and then implant the candidate TFT mutations). Placement of single mutations into test populations would also allow the unambiguous identification of the fertility-reducing mutation(s) harbored by the Brownsville haplotype, and to confirm whether it is indeed the

mt:Cyt-b mutation that causes the decrease in male fertility. If confirmed, the utility of the mt:Cyt-b mutation holds particular promise for pest control given that this candidate mutation has already been identified in a broad range of invertebrate and vertebrate species (*Clancy et al., 2011*).

Combined with previous studies, and a solid theoretical conceptual basis, our results lend credence to the utility of the TFT as a novel approach to pest control, deserving of continued development. Underpinning work needs to continue in the fruit fly model system to both explore the effects of linking multiple candidate TFT mutations within single mtDNA sequences (and quantifying their effects), and whether female-beneficial (but male fertility-impairing) haplotypes can be harnessed to drive the spread of TFT haplotypes through pest populations to effectively suppress population size. However, it would now also be timely to explore the capacity of TFT candidate mutations to decrease male fertility in other species, particularly real-world pest species that could be suitable targets for TFT control. If applicability and consistency of effect can be confirmed, the potential use of the TFT to suppress populations holds promise for a broad range of metazoan pests.

## Materials and methods

### Fly strains

The experiment harnessed a laboratory population of fruit flies that was originally collected in Dahomey (Benin, West Africa) in 1970 (*Partridge and Andrews, 1985*), and which has been kept at large effective population sizes since (at 25°C on a 12:12 light: dark cycle). To maintain the high levels of nuclear allelic variation segregating within the Dahomey population (*Gardner et al., 2005*; *Griffin et al., 2016*), populations have been kept in large replicate populations on a discrete-generations cycle since these were obtained from Prof Linda Partridge in 2010. This is achieved by propagating each generation with around 900 adult flies of 4 days of adult age, dispersed across three 250 ml bottles, each containing 75 ml of a potato-dextrose-agar food substrate. The flies are provided with a one to two hour ovipositing period, after which the number of eggs per bottle is manually reduced (trimmed) to 300–350. Adult flies are then removed from the bottles and then, for the subsequent generation, emerging adult offspring that eclose from each bottle are admixed prior to their re-sorting into three separate bottles to start the propagation procedure for the following generation.

We initiated six replicates of the Dahomey population, and introgressed the TFT mtDNA haplotype (sourced from Brownsville [BRO] Texas, USA; *Rand et al., 1994*) into three of these replicates. The other three replicates were designated to the control, and they hosted their own coevolved mtDNA haplotype sourced from Dahomey [DAH] (*Partridge and Andrews, 1985*). While the BRO haplotype confers low male fertility, with no recorded negative effects on female fertility, the DAH haplotype is putatively healthy and confers normal fertility in both sexes (*Camus et al., 2012*; *Yee et al., 2013*; *Wolff et al., 2016b*; *Dowling et al., 2015*; *Camus et al., 2015*).

All six replicates went through the same handling procedures leading into the experiment, over successive generations, which ensured effective population sizes across all replicates were carefully controlled with the expectation that levels of segregating nuclear variance were highly similar across replicates. Specifically, to initiate replicates harboring the TFT haplotype, 45 virgin females were collected from a genetic strain in which the BRO haplotype is placed alongside an isogenic nuclear background, called $w^{1118}$ (Bloomington #5905; *Ryder et al., 2004*). These females were then crossed to 50 males from the Dahomey lab population. In the next generation, 45 virgin daughters were collected from each strain replicate, and again backcrossed to 50 males from the Dahomey lab population. This backcrossing procedure progressed for 12 generations. To initiate replicates harbouring the DAH haplotype, the same procedure was followed, but the haplotypes were sourced directly from the Dahomey lab population, via an initial mating of 45 virgin females to 50 males collected from Dahomey. Then in the next generation, the virgin daughters of this cross were backcrossed to males of the stock Dahomey population, and this procedure repeated each generation (*Figure 4*).

Theoretically, each generation of introgression of the TFT haplotype into the Dahomey nuclear background increases the contribution of the Dahomey nuclear background by 50%, and thus, after 12 generations of introgression the contribution of Dahomey nuclear alleles to each TFT population replicate should have exceeded 99.98%. Thus, following the introgression procedure, we had six

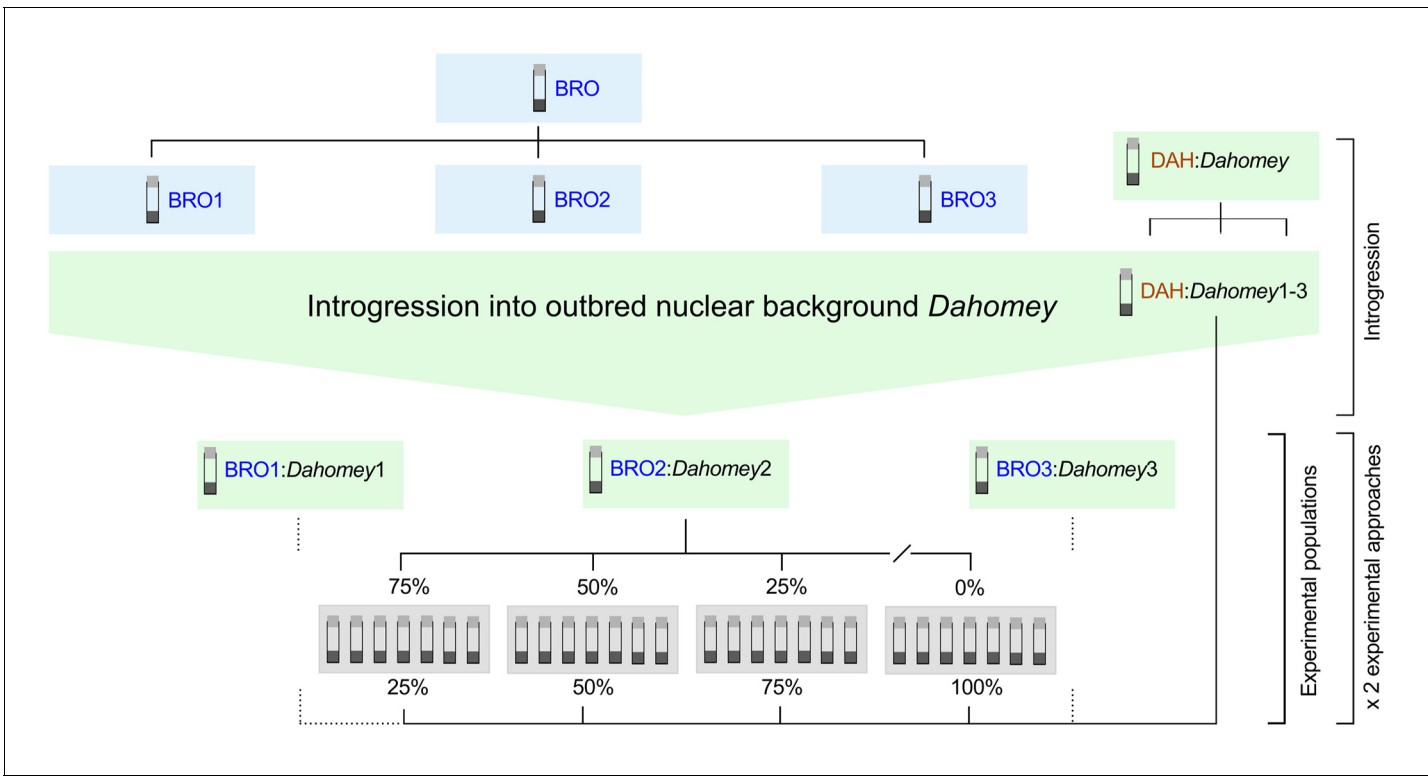

**Figure 4.** Experimental breeding scheme. The TFT-bearing BRO haplotype and the putatively healthy DAH haplotype were introgressed into the outbred nuclear background *Dahomey* in three independent replicates (i.e. BRO:*Dahomey*1-3; DAH:*Dahomey*1-3). Experimental populations were established by supplementing DAH:*Dahomey* test populations (DAH:*Dahomey*1-3) with varying contributions of fly pairs bearing the TFT haplotype (0%, 25%, 50%, 75%) sourced from the corresponding BRO:*Dahomey* replicate population (i.e. DAH:*Dahomey*1/BRO:*Dahomey*1; DAH:*Dahomey*2/BRO:*Dahomey*2; DAH:*Dahomey*3/BRO:*Dahomey*3). For each of the three biological replicates and for each treatment class (0%, 25%, 50%, 75%), we established seven technical replicate populations. These experimental populations were further duplicated, one cohort providing populations for the regulated population size approach (Experiment 1), and one cohort for the fluctuating population size approach (Experiment 2).

replicate strains, each of which contained a large representative sample of the nuclear alleles segregating within the Dahomey laboratory population; three of which however harbored the BRO haplotype harboring the candidate TFT mutations (denoted BRO/*Dahomey*), and the other three the DAH haplotype (denoted DAH/*Dahomey*).

All experimental crosses involved 4-day-old flies in 40 ml vials containing 6 ml of potato-dextrose-agar medium (at 25.0°C and a 12 hr: 12 hr light:dark cycle). Populations were maintained at a density of 80 flies per vial because at this density fly populations are sufficiently large to be stably maintained while limiting nutritional stress (e.g. food scarcity) which otherwise may impact development, fitness and behavior of affected fly populations (*Santos et al., 1994*). Further, sensitivity analyses within our a-priori demographic modeling, which informed our TFT treatments, showed that predictions were robust with respect to a modelled lab population size of 80 individuals (see Supplementary information in *Wolff et al., 2016b*).

## Experimental design

To test the capacity of the candidate TFT haplotype to effect population suppression within a multi-generational framework, in each experiment we established experimental target populations (DAH/*Dahomey*) that were seeded with varying contributions of the TFT haplotype-bearing BRO/*Dahomey* individuals (a TFT treatment with four levels). All populations were established with 80 adult fruit flies at 1:1 sex ratio, and with the TFT haplotype contributing to 0% (control), 25%, 50%, and 75% of the starting population. All four levels of the TFT treatment were established for each of the three biological BRO/*Dahomey* replicates, with each BRO/*Dahomey* replicate matched to a corresponding

DAH/*Dahomey* replicate. Within each strain replicate, each level of the TFT treatment was itself replicated seven times (i.e. three biological replicate BRO:*Dahomey* populations × four treatment levels × seven technical replicates = 84 experimental populations; see *Figure 4*).

We conducted two separate experiments, which ran concurrently. In the first experiment, we matched the conditions (in terms of density) under which fly populations are typically maintained in our laboratory, in which egg numbers per generation are carefully regulated, and thus, the population size is maintained around a constant density of 80 individuals at both juvenile and adult life stages (hereafter referred to as: *Experiment 1; Regulated* population size). Each experimental population was initiated using virgin flies at three days of age. The flies of each experimental population were then allowed 24 hr to mate, after which flies were transferred into vials with fresh food substrate for 4–6 hr for ovipositioning on day 4, until each vial contained in excess of 80 eggs. Immediately after ovipositioning, flies were collected and stored at −20°C, and the number of eggs per population reduced to 80 eggs by manually removing surplus eggs in each vial. To select eggs for retaining, the circular-shaped food source was divided into stripes. Eggs within each stripe were then counted starting from one side of the circle moving toward the opposite side of the circle until 80 eggs had been counted. This way, eggs were selected from the periphery through to the center of the food source regardless of egg density. Surplus eggs were discarded. We also aimed to minimize sampling effects by regulating egg density prior to the manual cull of eggs, by having females lay eggs over a short time period of only 4 to 6 hr. This ensured a sufficient number of eggs per vial, and also ensured that the majority of eggs contained in any one vial was used to give rise to the next generation. Each clutch of eggs was counted twice to minimize error in egg counts. Despite this precaution, eggs can be covered by food and escape detection, thus for 11 out of 840 vials (1.3%) offspring counts >80 were observed. The 80 eggs of each experimental population were then allowed to develop until eclosion to give rise to the next generation. Once eclosed into adults, flies of each population were transferred onto fresh food daily, until 4 days of age, when the next round of ovipositioning and egg-trimming occurred. We continued this procedure for 10 consecutive generations in total. In each generation, the total number of flies eclosing per population, from the initial pool of 80 eggs, was counted. The experiment was conducted blind to the identity of the experimental vials.

In the second experiment, all procedures were identical with the exception that the 84 experimental populations were not subjected to egg-trimming each generation (hereafter referred to as: *Experiment 2; Fluctuating* population size). Instead, once established, each ovipositioning period per experimental population was stopped when around 50% of the population vials were estimated to contain in excess of 80 eggs. Once this threshold was reached, flies were collected and stored at −20°C, eggs were left to develop to eclosion, and the number of eclosed flies per population in each generation was counted. This protocol diverges from the rearing conditions under which our laboratory populations are typically maintained, with the populations experiencing high levels of competition at both larval and adult stages for the 6 ml of available food, which routinely led to generations of population contractions, interspersed by generations of population expansions. This experimental design was used as an approximation for the demographic conditions natural populations may be exposed to, where single populations potentially transition through severe population bottlenecks, in which genetic drift would be expected to play a larger role in shaping frequencies of co-occurring mtDNA haplotypes relative to the regulated conditions of Experiment 1.

## Genotyping

DNA from single females was extracted in 96-well format using Wizard Genomic DNA Purification Kit (Promega, Madison, WI 53711, USA) following the manufacturer's instructions for single sample extractions, and using a third of recommended volumes to adjust for 96-well plate well volume. We invested most genotyping resources on Experiment 1, extracting DNA from seven females in each of three generations (1, 5, and 10) from each of the 63 populations that had been seeded with the TFT haplotype BRO (25%, 50%, and 75% BRO:*Dahomey* contribution; sample size: 63 populations × seven females × three generations = 1323 females).

For Experiment 2, we extracted DNA from nine females, all from generation 10 only, from each of the 63 populations that had been seeded with the TFT haplotype (25%, 50%, and 75% BRO:*Dahomey* contribution; sample size: 63 populations × nine females × one generation = 567 females). Populations established with DAH:*Dahomey* only (control populations) were not genotyped for

either experimental cohort (their genotypes were fixed at 0% TFT haplotype). All DNA extracts were adjusted to DNA concentrations of 5 ng*μl$^{-1}$ and a final volume of 50 μl per sample. Genotyping was conducted using a custom iPLEX Gold genotyping assay on the Sequenom MassARRAY Analyzer four system at Geneworks Pty Ltd, Thebarton, Australia.

## Statistical analysis

The structure of the data is outlined in *Figure 4*. We fitted linear mixed models to phenotype data (offspring number), and generalized linear mixed models to genotype data (frequency of the TFT haplotype), using the *lme4* package 1.1.12 (*Bates et al., 2015*) in R 3.0.3 (*R Development Core Team, 2013*).

In the phenotypic data analyses, the response variable (number of offspring produced) was modeled using a Gaussian distribution. Although this data is strictly count data, it conformed to a normal rather than Poisson distribution as expected of large sample sizes under the Central Limit Theorem. For example, the analysis of Experiment 1 contained only 1 zero value in the dataset. Although the analysis of Experiment 2 contained 15 zero values, removal of these values did not change the statistical inferences; and importantly, the residuals of these models were normally distributed. For the linear mixed models, fixed effects parameters were estimated using maximum likelihood estimation, and random effects were estimated using restricted maximum likelihood estimation. In the genotype data analyses, the frequency of the TFT haplotype was modeled as a binomial vector comprising the number of TFT haplotypes genotypes per vial and the number of wild-type haplotypes, using a binomial distribution and logit link.

For both analyses, we treated both the identity of the experimental population (i.e. individual vials) nested within Biological Replicate, and Biological Replicate, as random effects, and both TFT treatment (control, 25% TFT, 50% TFT, 75% TFT), and generation ($F_1$–$F_{10}$), as fixed effects. The TFT treatment control level was removed from analyses modeling the TFT haplotype frequency across the TFT treatments, since this level was invariably zero and its inclusion violated the model assumption of homogeneity of variance across classes. Generation was not added as a factor to the model of TFT haplotype frequency for Experiment 2, since genotyping was only conducted at generation 10 in this experiment. Consequently, each data-point represented that of a specific experimental population (i.e. 63 data points in the dataset, equaling 63 experimental populations), and thus, experimental population represented an observation-level effect in the models of Experiment 2. For the binomial models of TFT haplotype frequency, the *blmeco* package (*Korner-Nievergelt et al., 2015*) indicated overdispersion; 'experimental population' (an observation-level random effect) was thus added to the models to account for this.

We also examined the correlation between the number of offspring produced per experimental population and the TFT haplotype frequency, for both experiments. We fitted linear mixed models, with the number of offspring produced per experimental population as the response variable, and the TFT frequency of the same population as a fixed factor (of eight levels in Experiment 1, and 10 levels in Experiment 2), with the identity of the experimental population nested within the Biological Replicate, Biological Replicate, and generation number added as random effects to the model of Experiment 1, and with Biological Replicate added as a random effect for the model of Experiment 2.

Significance of fixed effects in each model was assessed using Type III sums-of-squares, χ2 tests in the *car* package of R (*Fox and Weisberg, 2011*).

## Acknowledgements

We thank Belinda Williams and Winston Yee for their help with fly husbandry, members of the Tompkins, Gemmell, and Dowling groups for helpful comments on the manuscript, and the three reviewers for their constructive comments and suggestions. This work was funded by a Smart Ideas grant from the New Zealand Ministry of Business, Innovation and Employment (MBIE).

## Additional information

### Funding

| Funder | Grant reference number | Author |
|---|---|---|
| Ministry of Business, Innovation and Employment | New Zealand, Smart Ideas Grant | Neil J Gemmell<br>Daniel M Tompkins<br>Damian K Dowling |

The funders had no role in study design, data collection and interpretation, or the decision to submit the work for publication.

### Author contributions

JNW, Data curation, Investigation, Visualization, Methodology, Writing—original draft, Writing—review and editing; NJG, Conceptualization, Funding acquisition, Writing—original draft, Writing—review and editing; DMT, DKD, Conceptualization, Formal analysis, Funding acquisition, Writing—original draft, Writing—review and editing

### Author ORCIDs

Jonci Nikolai Wolff, http://orcid.org/0000-0002-8809-5010
Neil J Gemmell, http://orcid.org/0000-0003-0671-3637
Damian K Dowling, http://orcid.org/0000-0003-2209-3458

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
