## [Decision Letter]

Thank you for submitting your article "Introduction of a male-harming mitochondrial mutation via Trojan Females achieves population suppression in fruit flies" for consideration by *eLife*. Your article has been reviewed by three peer reviewers, and the evaluation has been overseen by a Reviewing Editor and Diethard Tautz as the Senior Editor. The following individuals involved in review of your submission have agreed to reveal their identity: Liliana Milani (Reviewer #1) and Fabrizio Ghiselli (Reviewer #2).

The reviewers have discussed the reviews with one another and the Reviewing Editor has drafted this decision to help you prepare a revised submission.

Your manuscript describes the effects of a male-fertility impairing mtDNA haplotype on haplotype frequency change and population size variation in population cage experiments in the fruit fly *Drosophila melanogaster*. The experiments are well carried out and described. The three reviewers have provided constructive comments that critically evaluate your manuscript. The experiments seem to provide first support for the Trojan Female Technique. However, important issues have been identified by the reviewers that need to be addressed to fully prove this.

Important issues in this respect are:

1) Selection of 80 eggs and selection criteria of this upper limit

2) The limited effect (and replicability) and its implications for application

3) Identity of the mutation that caused the effect

Addressing these issues will be important in the light of a decision on your manuscript.

In addition, the reviewers have provided various additional valuable comments that deserved being addressed in a revision. Their comments are provided below.

*Reviewer #1:*

The work by Jonci N. Wolff, Neil J. Gemmell, Daniel M. Tompkins, Damian K Dowling, entitled "Introduction of a male-harming mitochondrial mutation via Trojan Females achieves population suppression in fruit flies", is the experimental evidence that the Trojan Female Technique (TFT) can be useful in population control as predicted by previous modelling (Gemmel. et al. 2013). Not only the TF mitochondrial mutation reduces male fertility, but there is no apparent pressure against the TFT haplotype considered (sometimes even showing increased frequency) that also appears to guarantee a persistent numerical suppression of target population.

The present work can be intended as a proof-of-concept for the use in TFT of the candidate mutation (mt:Cyt-b), that caused lower fertility for males (Dowling et al. 2015), in both competitive and noncompetitive mating contexts, across all nuclear backgrounds screened, with different genotype-by-environment interactions (Wolff et al. 2016).

I think the work is clear in describing the state of the art, the performed analyses, and the questions the new data are addressing. For these reasons, I recommend its publication.

Please clarify in the text what is meant for isogenic nuclear background.

Introduction, eighth paragraph: "Large and panmictic laboratory populations of *D. melanogaster* "

Discussion, fifth paragraph: "experimental populations utilized in the current study were panmictic”.

How can some sort of preferential mating be excluded?

Subsection “Experimental design”, second paragraph: I see from excel files that sometimes the offspring is of more than 80 individuals also in Experiment 1 (regulated population size). I imagine that the number of 80 eggs intended to be maintained had a certain error (and that it was not an error in the count of the offspring, that I imagine was of simpler calculation).

How were the eggs to be eliminated chosen? I am not an expert in *Drosophila* breeding, and I would like to know the best "random" way to reduce their number. From their different position in the vial? Were those closer to each other pruned? May the position of an egg be related in some way to its successful hatching, so a character to be considered?

Discussion, fifth paragraph: "[…] even though the experimental populations utilized in the current study were panmictic and characterized by high levels of segregating nuclear allelic variance".

How did the authors estimate the nuclear allelic variance of the used populations? I found no reference to this in Materials and methods when the introgression procedure was performed to obtain the 6 replicate strains.

What is the output of the genotyping analysis? Did you obtain both data on mitochondrial haplotype and nuclear alleles?

*Reviewer #2:*

The manuscript provides proof-of-concept for the Trojan Female Technique (TFT), a prospective population suppression technique based on naturally-occurring mitochondrial mutations that affect male fertility. TFT is a promising control strategy that could be applied to control pests.

The Introduction is clear, very well-written, and engaging. The experimental design is robust, and it required an impressive amount of work. Data collection and statistical analysis are appropriate. The authors discussed the results thoroughly.

Overall I really enjoyed reading the manuscript and in my opinion it represents an important study, very valuable for both applied (pest control) and basic research (mito-nuclear interaction and coevolution, effect of mitochondrial mutations).

I just would like to make a comment.

On the one hand, the authors observed that TFT-bearing BRO haplotype does not decrease in frequency through generations. This means that there is no direct evidence for selection against the TF haplotype. On the other hand, they also observed that the population suppression was below the expected levels in Experiment 1, and absent in Experiment 2.

I think that an explanation for this might be a sort of buffering effect consisting of "wild-type" haplotyes (DAH), complementing the suboptimal functionality of BRO haplotypes. Mitochondria carry multiple copies of mtDNA, and in order to have a phenotypic effect, a deleterious haplotype must be present in the mtDNA population of a mitochondrion at a frequency sufficient to counteract the complementing effect of functional haplotypes. The same rationale can be applied considering that there are multiple mitochondria in a cell, including of course germ cells, which are the main focus here (my collaborators and I discussed this topic extensively in [1] and [2]). Under this light, if a mitochondrial mutation is not severe enough to be purged by natural selection, it can persist in the population at a frequency that is determined by the severity itself, especially in this case where the mutation is deleterious only for males, but mtDNA has (almost always) matrilinear inheritance. That said, selection is still acting on male fertility, because each generation only the viable sperm will produce progeny, and because of this even if the suboptimal haplotype does not decrease in frequency, the observed population suppression is lower than expected (or absent).

An objection to this point could be about the TF haplotypes being segregated from "wild-type" haplotypes, but I think there are multiple ways by which an introgression can take place. Bottom line of this comment is: in my opinion, in the process of assessing the penetrance of a mitochondrial mutation, it is necessary to take into account the population dynamics also at mtDNA level (organelle, cell). I might be missing something in my considerations, so I would love to know alternative/complementary points of view.

In the future, it might be interesting to sequence samples from the populations included this study at different generations, to assess: 1) if compensatory mutations are arising in genes encoding subunits of the respiratory chain; 2) the frequency of TF and "wild-type" haplotypes, and if compensatory mutations are present also at the mtDNA level (e.g.: a mutation in cytb that compensates Ala278→Thr). Point 1) could be achieved by targeted deep sequencing of subunits of the electron transport chain (+ ATP syntase), and point 2) by deep sequencing of mtDNA.

Important note: I am including the following references only for the sake of scientific debate. I am *not* suggesting the authors to include such references in the Manuscript.

References:

1] Ghiselli, F. et al. Structure, transcription, and variability of metazoan mitochondrial genome: perspectives from an unusual mitochondrial inheritance system. Genome Biol. Evol. 5, 1535-1554 (2013).

2] Milani, L. & Ghiselli, F. Mitochondrial activity in gametes and transmission of viable mtDNA. Biol. Direct 10, 22 (2015).

*Reviewer #3:*

This is a well written manuscript that describes two experiments that test the effects of a male-fertility impairing mtDNA haplotype on haplotype frequency change and population size variation in population cage experiments in *Drosophila melanogaster*. The genotypes used have been previously described in other publications from the same group and that of David Clancy's group, namely a male-specific fertility defect in fruit flies harboring the Brownsville mtDNA haplotype. The main outcomes of this study show that under controlled densities, the proportion of Brownsville mtDNA haplotype is associated with a lowering of overall number of fruit flies (decreasing from 80) and this effect is numerically larger when the starting frequency of the impaired haplotype is higher. Overall, the suppression effect was a numerical reduction by ~8%. In contrast, in population cages that were allowed to fluctuate in size (around 80 flies), simulating a more natural population contraction-expansion, there was no suppression effect detected. The effects on haplotype frequency change were minimal throughout generational time, and the authors identified that heteroplasmy had established in some replicate cages.

I think the experiment was well conducted and the results well described but there are a few points that I feel require discussion or correction, outlined below.

Overall, the Introduction and Materials and methods are well described. One minor point – the reference to the fertility phenotype in the fifth paragraph of the Introduction refers to Yee et al. 2013 (male fertility), not Innocenti et al. 2011 (gene expression, which does not include any male fertility phenotypes).

One main point I have an issue with is the assertion that the defect is caused by the point mutation in the mtDNA CytB gene, which is repeatedly stated. While previous studies have suggested this association, there is no a priori reason to assume this mutation is any more important than the number of other mutations private to the Brownsville haplotype. For example, a study of mtDNA sequence variation from the same group (Wolff et al. 2015) provides sequence data across the whole mtDNA molecule and the regulatory D-loop alone harbors seven mutations that are private to Brownsville (D-loop alignment positions 303, 1466, 1469, 3244, 3250, 3251, 3709). Since these mutations are linked to the putative CytB mutation, there is insufficient evidence to suggest this mutation is the smoking gun. I think the use of safer description is warranted throughout and I suggest that the authors use the terms 'associated' and 'Brownsville haplotype' rather than the putative Ala278->Thr mutation.

It would be advantageous to know how much autosomal sequence variation there was in the Dahomey background used in this study (how outbred the lab-maintained stock really is, especially given the egg selection routinely used). Previous studies have shown near isogenic lines modify the effects of the Brownsville mtDNA, yet there is no assessment here; the Dahomey nuclear background also tends to exaggerate the Brownsville mtDNA haplotype effects. The magnitude of suppression effects is likely sensitive to nuclear background and only one was assessed here, therefore the generality of the finding is unknown. After all, Brownsville, TX, has a viable fruit fly population and this could form the basis of some discussion because it is the most 'natural' experiment. The Discussion, fifth paragraph, is unsubstantiated. Do you have any estimates of autosomal genetic diversity in these lines? If so, I would suggest including them, or at least discuss what may be expected in nuclear backgrounds that are not so sensitive to mtDNA effects, and which are likely to be experienced in a natural setting.

Is a -8% and stable population suppression biologically meaningful in the context of pest control, especially when it is only observed in a tightly regulated laboratory population? The Discussion mentions why the results may differ from the predicted values (and were somewhat less than expected) but the usefulness of the technique is somewhat limited (based on these findings). I think this could benefit from more discussion and especially for the role of egg-to-adult survival in the initial generations, which is known to be above average in Brownsville mtDNA lines.

Is the experiment #1 population size of 80 arbitrary, or was this based on the previous simulations? At such small population sizes, the opportunity for drift is very high and coupled with low and uneven sampling for the haplotype frequency estimates across both experiments, could this influence the results? Although there was vial replication, could these estimates not suffer from the same systematic bias? On the same note, the removal of flies from egg laying in experiment #2 (when approximately 50% of the vials produced 80 eggs) is still quite artificial and not what would be experienced in nature. This deserves some discussion since even a mild deviation from a strict egg laying dynamic can nullify the suppression effects found in experiment #1. It is perhaps unfortunate the reciprocal 100% Brownsville mtDNA treatment was not used as a control. While it would not be necessary or appropriate to use pest control in a population of zero flies, it would give a good estimate of the expected population size variation in the experimental design used here, which differs from previous studies using the Brownsville mtDNA haplotype. If these data are available, I would encourage the authors to include them to aid results interpretation.

The discussion of using genotypes with relatively high fitness females (Discussion, last paragraph) with low fitness in the corresponding males seems counterproductive since the object of pest control here is surely to reduce the number of insect vectors (not increase them!)

Perhaps wrap up with a more general commentary of the problems of evolution in gene drives as a pest control and how this technique can be tailored to avoid the same pitfalls – just a minor thought, but I feel this would compliment the nuclear compensation argument.

---

## [Author Response]

*Your manuscript describes the effects of a male-fertility impairing mtDNA haplotype on haplotype frequency change and population size variation in population cage experiments in the fruit fly Drosophila melanogaster. The experiments are well carried out and described. The three reviewers have provided constructive comments that critically evaluate your manuscript. The experiments seem to provide first support for the Trojan Female Technique. However, important issues have been identified by the reviewers that need to be addressed to fully prove this.*

*Important issues in this respect are:*

*1) Selection of 80 eggs and selection criteria of this upper limit*

Although, to the best of our knowledge, hatching success is not influenced by an egg’s position on the food source, we tried to keep egg selection as random as possible while remaining experimentally tractable. To select eggs for retaining, the circular-shaped food source was divided into stripes. Eggs within each stripe were then counted starting from one side of the circle moving towards the opposite side of the circle until 80 eggs had been counted. Surplus eggs were discarded. This way, eggs were selected from the periphery through to the center of the food source. We also aimed to minimize sampling effects by regulating egg density prior to the manual cull of eggs, by having females lay eggs over a short time period of only four to six hours. This ensured a sufficient number of eggs per vial, and also ensured that the majority of eggs contained in any one vial was used to give rise to the next generation. We have added further explanation of our process to Materials and methods (subsection “Experimental design”, second paragraph).

The upper limit of 80 eggs/larvae/individuals per vial was chosen because at this density fly populations are sufficiently large to be stably maintained while limiting nutritional stress (e.g. food scarcity) which otherwise may impact development, fitness and behavior. Accordingly, in the first experiment, all populations were started with 80 eggs in each generation. We now realize that we have failed to state this clearly, and have amended the manuscript to provide clarification in the Materials and methods section (subsection “Fly strains”, last paragraph). Further, sensitivity analyses within our a-priori demographic modelling, which informed our TFT treatments, showed that predictions were robust with respect to modelled lab population size of 80 individuals (now added as Supplementary Material).

*2) The limited effect (and replicability) and its implications for application*

In experimental categories where population size was affected by TFT treatment, effects were highly reproducible and were detected across all three strain replicates. In regard to the apparent lack of replicability across the two experiments (i.e. the suppression effect observed under regulated population dynamics not being observed under more stochastic dynamics), the lack of observable effect does not necessarily equate to a lack of phenotypic effect. Rather, the most probable explanation is that the TFT-mediated effects under the more stochastic dynamics in which large fluctuations in population size occur across single generations (populations experienced contractions and expansions amounting to on average 30% of their population size, per generation). This interpretation of the data is supported by the genotyping data, which showed an increase in TFT haplotype frequencies in Experiment 2 (more stochastic dynamics) for both the 25% and 50% treatments, and stable maintenance for the 75% treatment. We have amended the Discussion so that this observation is clearly articulated (Discussion, third paragraph).

The primary goal of the research presented was to provide proof-of-concept for the feasibility of using the TFT to suppress populations. This work thus presents an important step towards the implementation of the technique. As discussed in the manuscript, we acknowledge further development is required for its final and successful implementation in the field. While the 8% suppression effect we have observed in our experiments may appear modest, demographic modelling predicts that the same TFT effect will cause much greater suppression in natural populations where mating rates are likely to be much lower (details of the model used are now included with this paper, rather than referring to the Supplementary Material of a previous publication). The suggestion by the reviewer of inclusion of a discussion of the above-average pupal viability associated with the Brownsville haplotype is an excellent idea, and we have now incorporated this aspect in the Discussion (sixth paragraph). Ultimately, higher efficiencies of the TFT will likely be achievable via placement of several fertility-reducing mutations within a single TFT haplotype, or via the release of multiple TFT strains harbouring distinct TFT haplotypes. Since the writing of this manuscript another TFT mutation has been discovered independently by Patel and colleagues (Patelet al. 2016. A mitochondrial DNA hypomorph of cytochrome oxidase specifically impairs male fertility in *Drosophila melanogaster. eLife*, 5, e16923). This finding further supports the notion that animal mitochondrial genomes will be naturally enriched for male-harming mtDNA mutations, which can be harnessed to further increase the efficiency of the TFT. We have thus incorporated these new findings and elaborated on how these mutations can be harnessed to further develop the TFT (Discussion, seventh paragraph).

*3) Identity of the mutation that caused the effect*

*Addressing these issues will be important in the light of a decision on your manuscript.*

We agree with the reviewer that there is no definitive evidence that the mt:Cyt-b causes the male-sterility, and that additional mutations unique to the Brownsville haplotype located within the AT-rich region, may be responsible. The mt:Cyt-b has previously (and here) been highlighted, because it is the only mutation that causes a non-synonymous change. Following a classical genetics model, it is thus the only mutation with immediate consequences at the protein level in a functionally important enzyme complex. However, we acknowledge that this view is incomplete and perhaps too simplistic, and have incorporated the reviewer’s concern and recommendation into the manuscript. We have thus changed the wording throughout the manuscript, now referring to the *TFT haplotype* instead of the *mt:Cyt-b mutation*. We have further provided additional information in regard to the mutation harbored by the TFT haplotype, and in the Discussion we have further outlined that future experiments employing genome-editing technologies will be required to unambiguously single out the fertility-reducing mutation(s) harbored within the Brownsville haplotype (Introduction, sixth paragraph; Discussion, seventh paragraph).

*In addition, the reviewers have provided various additional valuable comments that deserved being addressed in a revision. Their comments are provided below.*

*Reviewer #1:*

*[…] Please clarify in the text what is meant for isogenic nuclear background.*

We have added a definition for isogenic nuclear background: “a nuclear background devoid of any allelic variation”.

*Introduction, eighth paragraph: "Large and panmictic laboratory populations of D. melanogaster"*

Discussion, fifth paragraph: "… experimental populations utilized in the current study were panmictic".

*How can some sort of preferential mating be excluded?*

We agree that there can still be mate preferences within our populations. But in this context, we use the word panmictic to indicate there are no genetic or behavioural restrictions between members of the Dahomey laboratory population (i.e. each fly can freely mate with other flies in this population – all individuals of the opposite sex in the population are potential partners, with no recorded reproductive isolation mechanisms causing divergence of the population). We are happy to remove the adjective panmictic to describe the population should the reviewer wish us to.

*Subsection “Experimental design”, second paragraph: I see from excel files that sometimes the offspring is of more than 80 individuals also in Experiment 1 (regulated population size). I imagine that the number of 80 eggs intended to be maintained had a certain error (and that it was not an error in the count of the offspring, that I imagine was of simpler calculation).*

We fully agree that a count of above 80 is most likely caused by human error during the egg-counting process. We tried to minimize error by counting each batch of eggs twice. However, eggs can be covered by food and escape detection. Values of 80 and above were recounted several times to exclude error in the offspring count. We have added an explanatory section to Materials and methods for clarification, and have also added information how often a count of 80 was observed (subsection “Experimental design”, second paragraph).

*How were the eggs to be eliminated chosen? I am not an expert in Drosophila breeding, and I would like to know the best "random" way to reduce their number. From their different position in the vial? Were those closer to each other pruned? May the position of an egg be related in some way to its successful hatching, so a character to be considered?*

We kept this process as random as possible, and experimentally feasible. To achieve this, we removed the circular-shaped food source from the lid (2 ml of food < 4mm thickness that sits in the lid of a cylindrical specimen container of 3 cm diameter), and divided this into stripes starting from one side of the circle moving towards the opposite side of the circle. Eggs were then manually counted, starting from one side of the food source, and moving towards the opposite side, until 80 eggs were counted. Surplus eggs were discarded. This way, eggs were chosen from all areas of the food source (e.g., from the periphery to the center) regardless of egg density. Furthermore, we aimed to minimize sampling effects by regulating egg density even prior to the manual cull of eggs, by having females lay eggs over a short time period of 4 to 6 hours). This ensured a sufficient number of eggs per vial, and also ensured that the majority of eggs contained in any one vial was used to give rise to the next generation, regardless of their position in the vial. To the best of our knowledge, hatching success is not influenced by an egg’s position on the food source. We have added further explanation to Materials and methods (subsection “Experimental design”, second paragraph).

Discussion, fifth paragraph: "[…] even though the experimental populations utilized in the current study were panmictic and characterized by high levels of segregating nuclear allelic variance".

How did the authors estimate the nuclear allelic variance of the used populations? I found no reference to this in Materials and methods when the introgression procedure was performed to obtain the 6 replicate strains.

*What is the output of the genotyping analysis? Did you obtain both data on mitochondrial haplotype and nuclear alleles?*

We have not directly measured the nuclear allelic variation maintained at specific genetic loci in our experimental populations. Rather, this inference is based on the results of quantitative genetic studies that routinely use the *D. melanogaster* Dahomey population used here, and have shown high amounts of additive genetic variance underpinning the expression of numerous life-history traits. See, for example:

Gardner, M.P., Fowler, K., Barton, N.H. & Partridge, L. (2005). Genetic Variation for Total Fitness in *Drosophila melanogaster.* Genetics, 169, 1553-1571.

Griffin, R.M., Schielzeth, H. & Friberg, U. (2016). Autosomal and X-Linked Additive Genetic Variation for Lifespan and Aging: Comparisons Within and Between the Sexes in *Drosophila melanogaster. G3: Genes|Genomes|Genetics*, 6, 3903-3911.

Since receiving the Dahomey population in our lab in 2010, we have kept it at very large effective population sizes (~900 adult flies per generation), to ensure that there is no bottleneck in allelic variation. We can thus be confident the experimental populations we have created harbor high levels of segregating nuclear variance. It was not our intention to suggest that we characterized the nuclear genetic variation in this experiment but we now see how current wording could have led to misunderstandings. We have thus amended that section for clarification, and have added relevant references (Discussion, fourth paragraph; subsection “Fly strains”, first paragraph).

*Reviewer #2:*

[…] I just would like to make a comment.

On the one hand, the authors observed that TFT-bearing BRO haplotype does not decrease in frequency through generations. This means that there is no direct evidence for selection against the TF haplotype. On the other hand, they also observed that the population suppression was below the expected levels in Experiment 1, and absent in Experiment 2.

*I think that an explanation for this might be a sort of buffering effect consisting of "wild-type" haplotyes (DAH), complementing the suboptimal functionality of BRO haplotypes. Mitochondria carry multiple copies of mtDNA, and in order to have a phenotypic effect, a deleterious haplotype must be present in the mtDNA population of a mitochondrion at a frequency sufficient to counteract the complementing effect of functional haplotypes. The same rationale can be applied considering that there are multiple mitochondria in a cell, including of course germ cells, which are the main focus here (my collaborators and I discussed this topic extensively in [1] and [2]). Under this light, if a mitochondrial mutation is not severe enough to be purged by natural selection, it can persist in the population at a frequency that is determined by the severity itself, especially in this case where the mutation is deleterious only for males, but mtDNA has (almost always) matrilinear inheritance. That said, selection is still acting on male fertility, because each generation only the viable sperm will produce progeny, and because of this even if the suboptimal haplotype does not decrease in frequency, the observed population suppression is lower than expected (or absent).*

*An objection to this point could be about the TF haplotypes being segregated from "wild-type" haplotypes, but I think there are multiple ways by which an introgression can take place. Bottom line of this comment is: in my opinion, in the process of assessing the penetrance of a mitochondrial mutation, it is necessary to take into account the population dynamics also at mtDNA level (organelle, cell). I might be missing something in my considerations, so I would love to know alternative/complementary points of view.*

*In the future, it might be interesting to sequence samples from the populations included this study at different generations, to assess: 1) if compensatory mutations are arising in genes encoding subunits of the respiratory chain; 2) the frequency of TF and "wild-type" haplotypes, and if compensatory mutations are present also at the mtDNA level (e.g.: a mutation in cytb that compensates Ala278→Thr). Point 1) could be achieved by targeted deep sequencing of subunits of the electron transport chain (+ ATP syntase), and point 2) by deep sequencing of mtDNA.*

*Important note: I am including the following references only for the sake of scientific debate. I am not suggesting the authors to include such references in the Manuscript.*

References:

*1] Ghiselli, F. et al. Structure, transcription, and variability of metazoan mitochondrial genome: perspectives from an unusual mitochondrial inheritance system. Genome Biol. Evol. 5, 1535-1554 (2013).*

*2] Milani, L. & Ghiselli, F. Mitochondrial activity in gametes and transmission of viable mtDNA. Biol. Direct 10, 22 (2015).*

We thank the reviewer for valuable comments, and welcome the reviewer’s thoughts on our results.Ultimately, we believe the modest levels of TFT-mediated population suppression in Experiment 1 were due to most females mating multiple times, with the negative effects of mating to a TFT male buffered by many females mating with both TFT and wild type males; thus leading to a compensation of reproductive success in the females. In Experiment 2, we observed no detectable population suppression at all, as the reviewer points out. We believe that the sheer magnitude of changes in population size between single generations (via population contraction and expansion, on average 30% per population per generation) in this second experiment were likely to have masked the more modest phenotypic effects (average of 8%) associated with the TFT mutation. We were able to show that despite strong population expansions and contractions, the introduced TFT haplotype was nonetheless stably maintained in most cases.

The reviewer suggests that these results could plausibly be explained by compensatory effects associated with each of the TFT mtDNA and wild type DNA segregating in heteroplasmy in different frequencies at the level of the organelle, cell and the individual; and/or the occurrence of compensatory modifier alleles either in the nuclear DNA sequence or the mtDNA. We agree with the reviewer that this explanation is hypothetically plausible, but believe that in our case, it is an unlikely scenario. When it comes to mitochondrial population dynamics, generally a mutant mtDNA molecule must reach between 60 and 80% frequency within a given tissue, relative to the wild type molecule, to exert its phenotypic effect, and this can lead to selection favouring shifts in the frequencies of mutant to wild type mtDNA molecules segregating within the cells of a single individual. As such, the reviewer suggests that future analyses of population dynamics of mtDNA variants would benefit from investigation across multiple levels – from organelle, to cellular, to individual. We agree that future analyses would benefit from deep-sequencing approaches that might be able to fully resolve the population dynamics of mtDNA evolution across organelle, cellular and individual levels. However, in our study, we detected occurrences of heteroplasmy in just a few cases, thus indicating that if heteroplasmy did commonly exist across our experimental flies, then it occurred at levels that were too low to be detected by conventional genotyping, and thus at levels that are unlikely to have yielded significant evolutionary implications (or implications for our experiment).

What about the interpretation that the modest effects of the TFT seen in Experiment 1, and the lack of effects in Experiment 2, could be caused by adaptive compensatory modifiers that have evolved either in the nuclear or mtDNA sequences? This interpretation has merit but, it is unlikely to explain our observed patterns (a suppression effect in Experiment 1, but not Experiment 2), given that the conditions under which Experiment 2 were run were much more restrictive than Experiment 1, in terms of the capacity for adaptive compensatory variants to evolve. That is, the population fluctuations that occurred throughout Experiment 2 would greatly magnify the influence of genetic drift in shaping allele frequencies across generations, relative to the conditions imposed in Experiment 1. Thus, compensatory adaptive variants should have been much less likely to evolve in Experiment 2 than Experiment 1. Although the overall TFT-mediated suppression effect is arguably modest in Experiment 1, we observed no clear signal of a compensatory response across 10 generations of the experiment (the 50 and 75% TFT treatments led to gradual reductions in population size with advancing generations, with no subsequent convergence, relative to the control treatment). Under a compensatory coevolution scenario, we would predict initial divergence between the control treatment (0% TFT) and the treatments with high starting frequencies of TFT variants, followed by a convergence, which would be indicative of adaptive compensation evolving as the experiment progressed across generations. In Experiment 2, we saw no such divergence in population sizes between the TFT and control treatment in the early generations of the experiment, suggesting there would never have been strong selection for a compensatory response.

We have currently not included this response in our revised manuscript as we believe its addition would distract from the discussion of our main finding. However, we would be happy to include our response if the editor feels that this addition would improve the manuscript.

*Reviewer #3:*

*[…] I think the experiment was well conducted and the results well described but there are a few points that I feel require discussion or correction, outlined below.*

*Overall, the Introduction and Materials and methods are well described. One minor point – the reference to the fertility phenotype in the fifth paragraph of the Introduction refers to Yee et al. 2013 (male fertility), not Innocenti et al. 2011 (gene expression, which does not include any male fertility phenotypes).*

We have corrected the reference from Innocenti et al. 2011 to Yee et al. 2013.

*One main point I have an issue with is the assertion that the defect is caused by the point mutation in the mtDNA CytB gene, which is repeatedly stated. While previous studies have suggested this association, there is no a priori reason to assume this mutation is any more important than the number of other mutations private to the Brownsville haplotype. For example, a study of mtDNA sequence variation from the same group (Wolff et al. 2015) provides sequence data across the whole mtDNA molecule and the regulatory D-loop alone harbors seven mutations that are private to Brownsville (D-loop alignment positions 303, 1466, 1469, 3244, 3250, 3251, 3709). Since these mutations are linked to the putative CytB mutation, there is insufficient evidence to suggest this mutation is the smoking gun. I think the use of safer description is warranted throughout and I suggest that the authors use the terms 'associated' and 'Brownsville haplotype' rather than the putative Ala278->Thr mutation.*

We agree with the reviewer that there is no definite evidence that the mt:Cyt-b causes male-sterility, and that additional mutations unique to the Brownsville haplotype located within the AT-rich region may cause the observed phenotype. The mt:Cyt-b has previously (and here) been highlighted as the putative candidate mutation because it is the only mutation that causes a non-synonymous change. Thus, following a classical genetics model, this mutation is the only mutation with immediate consequences at the protein level in a functionally important enzyme complex. However, we acknowledge that this view is incomplete and perhaps too simplistic, and have thus incorporated the reviewer’s concern/recommendation into the manuscript. We have changed the wording throughout the manuscript (including the title), now referring to the *TFT haplotype* instead of the *mt:Cyt-b mutation*. We have further provided additional information in regard to the mutation harbored by the TFT haplotype and, in the Discussion, we have further outlined that future experiments employing genome-editing technologies will be required to unambiguously single out the fertility-reducing mutation(s) harbored within the Brownsville haplotype (Introduction, sixth paragraph; Discussion, seventh paragraph).

*It would be advantageous to know how much autosomal sequence variation there was in the Dahomey background used in this study (how outbred the lab-maintained stock really is, especially given the egg selection routinely used). Previous studies have shown near isogenic lines modify the effects of the Brownsville mtDNA, yet there is no assessment here; the Dahomey nuclear background also tends to exaggerate the Brownsville mtDNA haplotype effects. The magnitude of suppression effects is likely sensitive to nuclear background and only one was assessed here, therefore the generality of the finding is unknown. After all, Brownsville, TX, has a viable fruit fly population and this could form the basis of some discussion because it is the most 'natural' experiment. The Discussion, fifth paragraph, is unsubstantiated. Do you have any estimates of autosomal genetic diversity in these lines? If so, I would suggest including them, or at least discuss what may be expected in nuclear backgrounds that are not so sensitive to mtDNA effects, and which are likely to be experienced in a natural setting.*

We agree that the magnitude of population suppression is sensitive to the nuclear background. We have previously examined the effect of the Brownsville haplotype on male fertility across several nuclear backgrounds and across different thermal environments. In previous experiments, we have shown that the Brownsville haplotype confers sub-fertility across a range of isogenic nuclear backgrounds; but also a range of “outbred” backgrounds in which the Brownsville haplotype had been placed inside of many flies, each fly of which contained a unique, but “population-representative” genotype that had been captured from one of three different lab-maintained populations (see Dowling et al. 2015, Evolutionary Applications). In these cases (where the Brownsville mtDNA was tested in an “outbred” background), we have verified that its fertility-impairing effect is general across 100s of different nuclear backgrounds. We also agree with the reviewer’s assessment that the Dahomey background tends to exaggerate the effect of the Brownsville haplotype relative to some other sample nuclear backgrounds (i.e. Coffs Harbour, LHM), but not relative to the Brownsville nuclear background or a nuclear background from Puerto Montt; also, note that we maintain backgrounds in our lab in which the Brownsville haplotype confers complete male-sterility.

In the current study we introgressed the Brownsville mtDNA haplotype into replicated outbred populations of the Dahomey laboratory population. As we have outlined above, in our response to reviewer 1, we did not directly measure the nuclear allelic variation maintained at specific genetic loci in our experimental populations. Rather, our inferences are based on the results of numerous quantitative genetic studies that routinely use the Dahomey population used here, and which have shown high amounts of additive genetic variance underpinning the expression of numerous life-history traits (e.g. Gardner et al. 2005 Genetics 169, 1553-1571; Griffin et al. 2016 G3 6, 3903-3911). Since receiving the Dahomey population in our lab in 2010, we have kept it at very large effective population sizes (about 900 adult flies per generation), to ensure that there is no bottleneck in allelic variation. When creating the six replicate Dahomey populations used in the current study (three of which harboured the BRO haplotype, and three of which harboured the DAH haplotype), we were careful to ensure that we did not put our populations through a bottleneck that would sizeably reduce levels of genetic variation. To this end, during the introgression procedure, we backcrossed 45 females of each replicate population to 50 males of the Dahomey laboratory population each generation. These numbers per population replicate are high when benchmarked against most studies utilizing experimental evolution approaches (e.g. artificial experiments in which an investigator seeks to maintain high levels of segregating genetic variance across numerous replicate population, while subjecting them to one of two divergent selection regimes; see Innocenti et al. 2014 BMC Evolutionary Biology 14:239; Bolstad et al. 2015 PNAS, 112-13284-13289. Thus, our approach is gold-standard from an experimental evolution perspective, and utilizes a well-studied laboratory population of *Drosophila* that has previously provided numerous insights into the adaptive capacity of populations. We are confident the experimental populations we have created each harbour high levels of segregating nuclear variance that are all highly representative of levels in the stock Dahomey population we continue to maintain in our laboratory.

Finally, it was not our intention to suggest that we characterized the nuclear genetic variation maintained in the Dahomey population in this experiment, but we see how the wording in the previous version of the manuscript could have led to misunderstandings. We have incorporated the additional information regarding the Dahomey nuclear background (Discussion, fourth paragraph; Subsection “Fly strains”, first paragraph), and suggestions regarding the incorporation of additional backgrounds, into the discussion (Discussion, fourth paragraph).

*Is a -8% and stable population suppression biologically meaningful in the context of pest control, especially when it is only observed in a tightly regulated laboratory population? The Discussion mentions why the results may differ from the predicted values (and were somewhat less than expected) but the usefulness of the technique is somewhat limited (based on these findings). I think this could benefit from more discussion and especially for the role of egg-to-adult survival in the initial generations, which is known to be above average in Brownsville mtDNA lines.*

The primary goal of the research presented here was to provide proof-of-concept for the feasibility of using the TFT to suppress population size. We achieve 8% suppression under the conditions of Experiment 1. Nonetheless, as noted by the reviewer, this effect was lower than what we had been expecting. However, our a-priori demographic modelling predicts that the detected effects in our study are likely to underrepresent the likely efficacy of the TFT in wild populations where mating rates are likely to be much lower. The inclusion of the above-average pupal viability for the Brownsville haplotype is an excellent suggestion for further discussion, and we have now incorporated this into the manuscript (Discussion, sixth paragraph).

We also note that higher efficiencies in future development of the TFT are likely to be achievable via the combination of several fertility-reducing mutations within a single TFT haplotype, or via the release of multiple TFT strains harbouring distinct TFT haplotypes. Since the writing of this manuscript, another TFT mutation has been discovered independently by Patel and colleagues (Patelet al. 2016. A mitochondrial DNA hypomorph of cytochrome oxidase specifically impairs male fertility in *Drosophila melanogaster. eLife*, 5, e16923). This finding provides support for the notion that animal mitochondrial genomes should be naturally enriched for male-harming mtDNA mutations, and which can be harnessed to further increase the efficiency of the TFT. We have thus incorporated these new findings and elaborated on how these mutations can be harnessed to further develop the TFT. We have also incorporated the reviewer’s suggestion relating to multiple-release strategies (Discussion, seventh paragraph).

*Is the experiment #1 population size of 80 arbitrary, or was this based on the previous simulations? At such small population sizes, the opportunity for drift is very high and coupled with low and uneven sampling for the haplotype frequency estimates across both experiments, could this influence the results? Although there was vial replication, could these estimates not suffer from the same systematic bias? On the same note, the removal of flies from egg laying in experiment #2 (when approximately 50% of the vials produced 80 eggs) is still quite artificial and not what would be experienced in nature. This deserves some discussion since even a mild deviation from a strict egg laying dynamic can nullify the suppression effects found in experiment #1. It is perhaps unfortunate the reciprocal 100% Brownsville mtDNA treatment was not used as a control. While it would not be necessary or appropriate to use pest control in a population of zero flies, it would give a good estimate of the expected population size variation in the experimental design used here, which differs from previous studies using the Brownsville mtDNA haplotype. If these data are available, I would encourage the authors to include them to aid results interpretation.*

The upper limit of 80 eggs/larvae/individuals per vial was chosen because at this density fly populations are sufficiently large to be stably maintained while limiting nutritional stress (e.g. food scarcity) which otherwise may impact development, fitness and behavior. Accordingly, in the first experiment, all populations were started with 80 eggs in each generation.

We now realize that we have failed to state this clearly, and have amended the manuscript to provide clarification in the Materials and methods section (subsection “Fly strains”, last paragraph).

Our a-priori demographic modelling, which informed our TFT treatments, was parameterised based on a population size of 140. However, sensitivity analyses showed that predictions were robust with respect to modelled lab population size of 80 individuals. We have added this additional information as supplementary material. As we have argued in our response to the previous comment, sample sizes of ~ 40 breeding pairs represent the gold standard in experimental evolution studies of invertebrates. Since egg-to-adult viability rates are high in *D. melanogaster*, we expect that the effective population sizes arising from the 80 eggs per generation in Experiment 1 were high, and thus the efficacy of natural selection would have greatly outweighed the effects of drift. The fact that our high TFT treatments steadily diverged from the other treatments, and that this was maintained across the 10 generations of the trial, suggests that genetic drift was not the predominant driver of haplotype frequencies per vial. Critical to this discussion, however, is the fact that we had replicated each of our treatments across three strains, and within each strain we had 7 experimental populations. Thus, each treatment was effectively replicated 21 times. Because the effects we detected were consistent (i.e. the effects detected statistically were strong, and the patterns were upheld across vials), this confirms that the haplotype frequencies across vials were not determined by drift (since this would have led to random haplotype frequencies across vials across the treatments – detectable via large vial-specific or replicate-specific effects, and no TFT-mediated effect). We can confidently conclude that the results of Experiment 1 are not affected by genetic drift.

We also agree that stopping ovipositioning in Experiment 2 when around 50% of the population vials contained in excess of 80 eggs may be a simplistic attempt to imitate the complex population dynamics of natural populations. However, this experimental design was used as an approximation for the demographic conditions natural populations may be exposed to in terms of contractions and expansions of population size, where populations potentially transition through severe population bottlenecks which may affect the frequencies of co-occurring mtDNA haplotypes via drift.. We have highlighted our intention and associated limitations in the manuscript (subsection “Experimental design”, last paragraph). Thus, in contrast to Experiment 1, haplotype frequencies in experimental populations of Experiment 2 were much more likely to be affected by drift. Remarkably, however, while we did not detect suppression of population sizes at the higher TFT treatments in this second experiment, the frequencies of the TFT haplotypes were nonetheless stably maintained in the 50 and 75% treatments; thus suggesting that even under these conditions, drift was not the primary determinant of the competing mtDNA haplotypes.

While we agree with the reviewer that adding a 100% TFT treatment would be interesting, we were under heavy logistical constraints since we were already maintaining 164 experimental populations, and were working at the upper level of what was possible in the lab. We thought it much more valuable to have the 0% control, with which we could compare the effects of the other TFT-treatments to. The 100% treatment would have been informative, since it would have led to the greatest selection for compensatory adaptations (given that all males would have been carrying the male fertility-reducing haplotype). These data are not available however. We have added a section to highlight that inclusion of this approach may aid the interpretation in future experiments (Discussion, fourth paragraph).

*The discussion of using genotypes with relatively high fitness females (Discussion, last paragraph) with low fitness in the corresponding males seems counterproductive since the object of pest control here is surely to reduce the number of insect vectors (not increase them!)*

We see the reviewer’s point, and have changed the wording in this section to clarify that population suppression is the ultimate goal (Discussion, last paragraph). However, a mutation that confers high female fertility, but has devastating effects on male fertility, is potentially the perfect TFT mutation because the high female fertility will act to “drive” the TFT mutation into the pest population. Once the TFT mutation is at high frequencies, most males would have reduced fertility, and populations would crash.

*Perhaps wrap up with a more general commentary of the problems of evolution in gene drives as a pest control and how this technique can be tailored to avoid the same pitfalls – just a minor thought, but I feel this would compliment the nuclear compensation argument.*

We agree, and have added a section to the Discussion, highlighting/discussing the problem of the emergence of resistance alleles in response to both the TFT and gene-drive systems (Discussion, fourth paragraph).